# Assessing the Subjective Effectiveness of Sensorimotor Insoles (SMIs) in Reducing Pain: A Descriptive Multicenter Pilot Study

**DOI:** 10.3390/jfmk8020066

**Published:** 2023-05-18

**Authors:** Stephan Becker, Steven Simon, Jan Mühlen, Carlo Dindorf, Michael Fröhlich

**Affiliations:** 1Department of Sport Science, RPTU Kaiserslautern-Landau, 67663 Kaiserslautern, Germany; 2Department of Molecular and Cellular Sports Medicine, Institute of Cardiovascular Research and Sports Medicine, German Sport University Cologne, 50933 Cologne, Germany

**Keywords:** insole, proprioception, conservative treatment, gait analysis, kinesiology, sensorimotor foot orthosis, orthopedic footwear

## Abstract

This pilot study aimed to investigate the use of sensorimotor insoles in pain reduction, different orthopedic indications, and the wearing duration effects on the development of pain. Three hundred and forty patients were asked about their pain perception using a visual analog scale (VAS) in a pre–post analysis. Three main intervention durations were defined: VAS_post: up to 3 months, 3 to 6 months, and more than 6 months. The results show significant differences for the within-subject factor “time of measurement”, as well as for the between-subject factor indication (*p* < 0.001) and worn duration (*p* < 0.001). No interaction was found between indication and time of measurements (model A) or between worn duration and time of measurements (model B). The results of this pilot study must be cautiously and critically interpreted, but may support the hypothesis that sensorimotor insoles could be a helpful tool for subjective pain reduction. The missing control group and the lack of confounding variables such as methodological weaknesses, natural healing processes, and complementary therapies must be taken into account. Based on these experiences and findings, a RCT and systematic review will follow.

## 1. Introduction

Biomechanical insoles are an established component of healthcare systems as a therapeutic intervention, whereas therapeutic evidence for sensorimotor—also known as proprioceptive or afferent stimulation—insoles (SMIs) is still unclear, and they are often not subsidized by healthcare systems and health insurance companies, citing a lack of evidence. For a better understanding of this comparatively new form of therapy with a low amount of evidence, first, their difference compared to biomechanical insoles is determined, the concept is explained, and the existing evidence for SMIs is presented for clarity.

Biomechanical insoles have been used in the context of orthopedic problems for decades, often as a standalone treatment or in addition to medication and/or active therapy, depending on foot [1,2,3,4], knee [5,6], hip [7,8], and back indications [5]. Insoles are part of both cause and symptom management. Postural control and gait pattern are influenced by the cutaneous sensitivity of the plantar foot surface, providing a starting point for orthopedic therapy [9,10,11]. This plantar sensitivity is assumed to be influenced by intervening in the fundamental transduction and transmission of tactile feedback of 1702 detected cutaneous mechanoreceptors and their reflexive zones; e.g., using insoles [12]. Here, a technical distinction can be made in orthopedic footwear between primarily biomechanical and sensorimotor [13,14,15]. The use of sensorimotor—also known as proprioceptive or afferent stimulating—insoles (SMIs) as an alternative approach to primarily biomechanically supporting insoles has been gaining increasing importance in the context of orthopedic patient shoe care for many years [13,16,17,18]. The essential difference of the primarily biomechanically acting insoles, which work via passively supporting, bedding, and pressure redistributing elements, is that the body’s own regulatory systems are actively influenced and utilized via the proprioceptors (muscle spindles, Golgi tendon apparatuses, Vater–Pacini corpuscles) in SMI by means of precisely defined pressure points [19,20,21]. In the scientific literature and in orthopedic shoe care practice, different SMI concepts can be found, including Jahrling, Derks, Pfaff, Woltring/Springer, or Sensoped. Jahrling’s and Woltring/Springer’s show the most objective evidence, in addition to a transparent presentation of functional effects [22].

Based on the goal of generating afferent stimulation using individual pelotte or protrusion placement [22], the following mechanisms are considered:Muscle tonus reduction: Lengthening of muscle tendons produces tonus reduction of a muscle [13,19,22]. The application of a pressure point to the tendon lengthens and therefore stretches it. Through the Golgi tendon reflex, a tonus reduction in the inserting muscle occurs as a physiological response. In particular, the toe flexors and the plantar fascia respond to this mechanism.Muscle tonus increase: A pressure spot produces a directional force with a shortening effect of muscle tendons, which results in an increase of the muscle tonus [13,19,22]. Approaching the muscle tendon insertion to its origin neurobiologically triggers muscle spindles, and thus initiates sensitively increased muscle activity for an appropriate muscle tonus. The reduced tendon tonus is readjusted by the muscle to the required length. Type 2 afferents may also be responsible for a short-term increase in activity [17]. This mechanism is used for M. peroneus longus [17,18] and M. tibialis posterior to increase their muscle activity and support a foot supination or pronation depending on the overall foot kinematics.Indirect mechanical effects: Elements of the insoles can result in a rightening up of a structure. Calcaneus stabilization may have a beneficial effect in the case of excessive internal rotation of the leg by suppressing pronation motion of the subtalar joint [23]. The foot and leg axis and the center of pressure move away from a stimulating element that, as a result, can be used to influence internal or external foot rotation.

It seems undisputed that changes in the surface of the insole can lead to changes in muscle activity and consequently affect the motor system [10,24,25]. Tonus increasing or decreasing modulations based on the individual foot pattern, and thus the active stimulation of the musculature [20,26], are expected to provide additional treatment benefits in the context of treatment with SMIs [27]. Previous research supports this concept [18,20,26,28,29], but further perspectives and investigations are required [30].

The aim of this study was to investigate the unaddressed question of the influence of SMI on the subjective perception of pain, in addition to examining the possible effects of indication and worn duration. The full effect of SMI is yet to be proven. A change in investigation perspective, in comparison to previous studies, is needed. There have been no studies published that examine the effectiveness of SMI regarding the subjective perception of pain. In addition to objective evidence of effectiveness, subjective perception also plays a key role. After all, achieving pain relief is and remains a primary therapeutic goal.

## 2. Materials and Methods

### 2.1. Patients and Insoles

The sample size was calculated prior, using G*Power (Version 3.1.9.6 for Macintosh, University of Kiel, Kiel, Germany) for a repeated ANOVA (between factors, f = 0.25, α = 0.05). A minimum group size of 212 persons was calculated (power 0.952), which was increased due to expected dropouts. A total of 340 persons (197 women, 143 men) completed the study and were included (see Table 1). There were 19 dropouts (e.g., missing retest), who were excluded from the sample size beforehand. All participants were under medical treatment for a specific indication (foot, Achilles tendon, knee, back), but had no other complaints. The study was based on and carried out in accordance with the current guidelines of the Declaration of Helsinki and was approved by the responsible ethics commission (Ethikkommission RPTU Kaiserslautern-Landau, Fachbereich Sozialwissenschaften, Nr. 55, 12/2022 approved 14 December 2022). All participants signed informed consent forms, which included granting permission to publish the results. The authors have no conflicts of interest to declare.

Eighteen German medical supply stores with certified orthopedic shoe technicians participated in this study and were commissioned to manufacture SMIs. The subjects were individually fitted with SMIs (see Figure 1) according to the Woltring/Springer system (Springer Aktiv AG, Berlin, Germany) based on their complaints in different body areas.

### 2.2. Methods

An 11-item visual analog scale (VAS) [31] was used to assess subjective pain perception before (VAS_pre) fitting the insoles and after (VAS_post) a specific worn duration. In each case, the VAS query refers to the personal indications (foot, Achilles tendon, knee, back) that were the reason for the doctor’s consultation and the insole treatment. The timing of VAS_post was based on individual follow-up appointments scheduled by the orthopedist and orthopedic shoe technician, but no earlier than 6 weeks after fitting. Consequently, the intervention duration was divided into three categories: less than 3 months, 3 to 6 months, and more than 6 months.

Within this time, the insoles were worn for at least one to two hours per day during a two-week familiarization period, followed by full-day wear during the intervention period. In case of any discomfort, the SMI was adjusted. All participants were provided with verbal and written instructions for full-day wear: After the familiarization period (1–2 h/day), the SMIs must be worn any time the participant wears shoes. In case the SMI does not fit a particular shoe, this shoe should be avoided to ensure therapy and evaluation. Participants confirmed this during VAS_post.

To avoid small subgroups in the present sample, two repeated-measure ANOVAs were separately calculated for the between-subject factors “indication” (model A) (foot, Achilles tendon, knee, back) and “worn duration” (model B) (up to 3 months, 3 to 6 months, more than 6 months), each with the within-subject factor “time of measurement”. The information regarding the indication was missing for 7 subjects and the worn duration was missing for 3 subjects. To keep the number of subjects for the calculations as large as possible, only the subjects with missing information mapping the between-subject factors of interest for each individual repeated ANOVA were excluded, resulting in a total of 333 subjects for model A and for 337 for model B.

Since the manual examination of outliers exceeding 1.5 times the interquartile range confirmed that these were not measurement errors but actual extreme values, they were not removed. Due to the presence of these outliers, as well as partial inhomogeneity of the covariance after the box test (*p* < 0.01), the assumptions for calculating a commonly used mixed ANOVA could not be met. Therefore, a robust mixed ANOVA using 20% trimmed means was used for statistical analysis with the R package WRS2 (Mair and Wilcox, 2020). Visualizations were performed using the Python library “Seaborn” [32].

## 3. Results

The results show significant differences between the within-subject factor “time of measurement” for both ANOVAs (model A: F(1, 126.77) = 1130.05, *p* < 0.001, np_2_ = 0.90; model B: F(1, 16.64) = 454.66, *p* < 0.001, np_2_ = 0.97).

Figure 2 shows the corresponding changes in the rating of pain between the times of measurements. The very large effect size is also visible, evidenced by the slight overlap of the distributions of the two measurement times in the figure. Only a few subjects showed a deterioration over the measurement times.

No interaction of indication and points of measurements (model A) was present (F(3, 47.32) = 0.04, *p* = 0.99). Furthermore, no interaction between worn duration and points of measurements (model B) was found (F(2, 15.42) = 0.81, *p* = 0.46).

## 4. Discussion

The results show that subjective pain perception was reduced by a mean of 5.04 on the VAS (*p* < 0.001), which can be considered a positive primary treatment goal achievement and in line with findings by Wegener, Wegener, Smith, Schott, and Burns [30] concerning the comfort and cushioning of SMIs. The positive effects of SMIs on muscular activity, joint kinematics of the foot and knee during gait, and various postural parameters have already been measured in several studies [18,20,26,28,29,33]. These improvements may in turn be the reason for a reduction in the nonspecific pain of the subjects. However, the authors are only aware of studies investigating the influence of biomechanical insoles on pain perception [34,35]. Both studies showed a significant pain reduction.

An essential therapeutic objective of SMIs, which could be one factor for the reduced pain perception in this study, is the activation of M. peroneus longus and M. tibialis posterior that stabilize the ankle joint. As a result, the corrected foot position and a better foot pressure distribution can lead to a reduction in foot pain. This stabilization and readjustment of the joint kinematics would also explain the reduced discomfort in other joint segments due to biomechanical linkage [36,37]. Nevertheless, a change in foot position and loading can also have contrary effects in the sense of a greater sensation of discomfort or even pain caused by the changed joint kinematics [11]. This must always be taken into account by the orthopedic shoe technician. Schmitt et al. [26] were able to measure an activating effect of SMIs in soldiers with regard to M. peroneus longus, but not to M. tibialis anterior. M. tibialis anterior is used in surface electromyography (sEMG) because of its synergistic function in supporting midfoot supination and rearfoot inversion [37]. However, SMIs target the M. tibialis posterior, which cannot be derived using sEMG. Ludwig, Quadflieg, and Koch [17] obtained comparable results in a blinded crossover study. When wearing the Springer/Woltring SMIs, the activity of M. peroneus longus increased significantly during the stance phase compared to dummy insoles. This increase in activity may be used to reduce rearfoot inversion or midfoot supination, which may have a positive effect on unfavorable joint kinematics and, as a consequence, on the subjective pain perception of patients with insole-indicated complaints.

In the case of Achilles tendon complaints, the focus is on the elements that have a tonus-reducing effect on the plantar fascia and plantar flexors of the upper ankle joint (M. gastrocnemius, M. soleus). By tonus reduction of the calf muscles, for example, an Achilles tendonitis should heal better. However, there is still a lack of scientific data to support this approach. In the context of knee and back complaints, the orthotist must consider the individual biomechanical coupling of the foot position and its kinematic effects to the higher joint system, both in stance and in motion. It is not possible to standardize the use of insoles for the same indication. The decision regarding the necessary design of SMI elements is made by the orthotist or attending physician. In contrast to previous laboratory investigations, the present study provides a perspective on everyday life conditions.

While it seems plausible that improving the foot position with insoles may contribute to foot therapy and Achilles tendon complaints, it is still unclear to what extent insoles may contribute to the improvement of nonspecific knee and back pain based on biomechanical and sensorimotor effects. Ohlendorf et al. [38] showed that wearing SMIs improved upper body posture and, in line with the pain development of this sample, highlighted that they helped reduce subjective postural discomfort ratings. Similarly, Dankerl, Keller, Häberle, Stumptner, Pfaff, Uder, and Forst [28] showed that SMIs contributed to postural changes. This may be an influencing factor in why participants in the present study reported a reduction in back discomfort as a result of wearing SMIs. Sensory information provided by muscle and skin afferents is important for upright posture and balance control [39]. Insoles have been used as a strategy to improve the quality of cutaneous feedback [40]. In particular, the toes and lateral areas of the plantar surface are important regions for balance control [41]. According to Almeida et al. [42], postural insoles improve the alignment of the knees, hips, pelvis, and spine by adjusting the distribution of plantar loads from initial contact to the ground up to mid stance phase. One model dictates that the correction of increased calcaneal eversion can reduce inward rotation of the tibia, fibula, and femur, thereby reducing pelvic anteversion [43]. Reducing increased pelvic tilt may, in turn, have a positive effect on lumbar hyperlordosis, which is a potential cause of lower back pain [44]. However, the overall evidence for the effects of SMIs in general and, specifically, for the treatment and prevention of back pain is weak [5]; however, there are plausible biomechanical linkages.

In general, when interpreting the results, it should be taken into account that natural and autonomous healing processes of the human body might be influencing factors in pain relief (temporal effect). This means that an improvement in symptoms might be possible even without wearing insoles (e.g., rest and avoidance of sports). Other therapeutic treatments (e.g., manual therapy, thermotherapy, shockwave therapy) may also have contributed to pain relief during the intervention period. For example, in the case of back pain, a physician usually prescribes further conservative therapy, such as physiotherapy. Anti-inflammatory medication may also have had similar beneficial effects. In addition, some data show a worsening of the pain pattern. Missing information regarding the influence of other therapies, or even the absence of therapies that may have been necessary in addition to the insole treatment, could be reasonable explanations. The lack of information on concomitant therapies must be considered as a clear limitation in the interpretation of the results.

Furthermore, no diary was kept on the daily worn duration of the insoles. However, this would be quite important with regard to the assessment of the influence of the deposits and their possible effect. It also must be considered that pain is not only one-dimensional, as represented by the VAS, but multidimensional in its genesis.

Finally, the results showed that there were no significant differences between the different worn duration groups. The overall worn duration was consistent with the individualized recommendations of the physician and orthopedist who fabricated the SMIs. This corresponds to typical insole therapy practice. For scientific purposes, a clearer standardization with multiple measurement points is needed to investigate the valid influence of worn duration.

### 4.1. Future Studies and Strengths of This Study

Future studies in conditions of everyday life should include detailed initial, follow-up, and baseline examinations, as well as a comprehensive and validated questionnaire and medical imaging techniques that include additional pathologic, physiologic, and anthropometric parameters, including detailed foot information, such as the arch index [45], to calculate the effects of SMIs with respect to the foot shape of the included sample (e.g., hollow foot or flat foot). The inclusion of a control group after randomization is also recommended, which should be fitted with primarily supportive foot orthoses. Ideally, a placebo group should be added. In this way, the effect of SMIs can be observed in a more differentiated manner over different worn durations. In addition, on the basis of the next randomized controlled trials (RCT), gender and age effects should be investigated.

The strengths of this multicenter study can be seen in the fact that all previous investigations on SMIs have been conducted based on kinetic or electromyographic parameters and laboratory settings. None of them have asked for the subjective impression of pain development from using SMIs daily. The subjective perception of patients is a key factor in many fields and should not be underestimated [46,47].

Furthermore, the high sample size should certainly be mentioned. On the basis of these results, further research (e.g., RCTs) can be planned.

### 4.2. Limitations

Nonetheless, there are strong limitations of this descriptive pilot study that must be considered for interpretation and following studies. First of all, it should be mentioned that, due to the nature of this pilot study, no control group is available for comparison. The present one-arm design strongly limits (e.g., placebo effect) the generalizability of the results. Additionally, a comparison to a group with biomechanical insoles to control possible selection bias is missing. Furthermore, potential confounding factors in the form of accompanying therapeutic treatments were not controlled. The natural healing process (temporal effect) may also be important; i.e., the pain perception could have improved on its own, if the person took more care of themselves. Moreover, it should be mentioned that pain is not only one-dimensional, as represented by the VAS, but also multidimensional. For this reason, pain should be multidimensionally measured to allow for a precise understanding of pain and its development in future studies. Ultimately, therefore, the individual worn duration must be better quantified in follow-up studies in order to be able to better assess the impact of the insoles.

## 5. Conclusions

The present multicenter pilot study on SMIs must be cautiously and critically interpreted. The results show a significant subjective reduction in subjective pain over different worn durations and indications (foot, Achilles tendon, knee, back). However, a strong statement on the effectiveness of SMIs on the reduction in subjective pain perception cannot yet be made on this basis, although there are plausible explanations and supporting results from other studies, and the primary therapeutic goal of pain reduction was achieved. The knowledge and experience gained from this descriptive pilot study on the possible reduction in pain through the use of sensorimotor insoles justifies further investigations in the form of a systematic review and a RCT.

## Figures and Tables

**Figure 1 jfmk-08-00066-f001:**
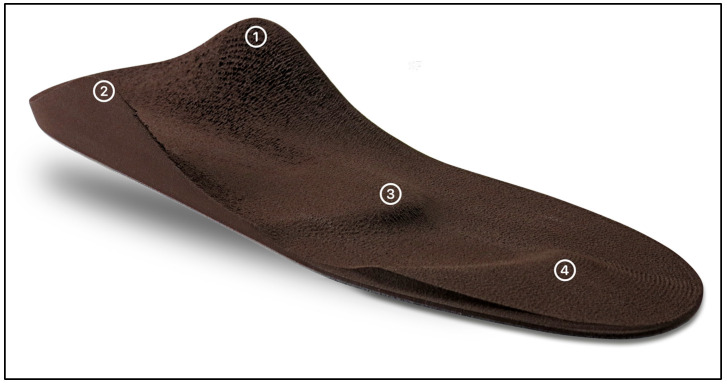
SMI according to Woltring/Springer with a strong profiling on the plantar side (characterized by four essential elements: Element 1: medial rearfoot spot with a height of approximately 20 mm in the area of the sustentaculum tali (postulated function: increasing the tonus of the M. tibialis posterior muscle); Element 2: lateral rearfoot spot (postulated function: increasing the tonus of the M. peroneus longus muscle); Element 3: retrocapital pad (postulated function: decreasing tonus of the plantar fascia and the plantar flexors of the foot); Element 4: toe bar (postulated function: pre-stretching of the toe-flexors and promoting postural control through stimulation of the mechanoreceptors)).

**Figure 2 jfmk-08-00066-f002:**
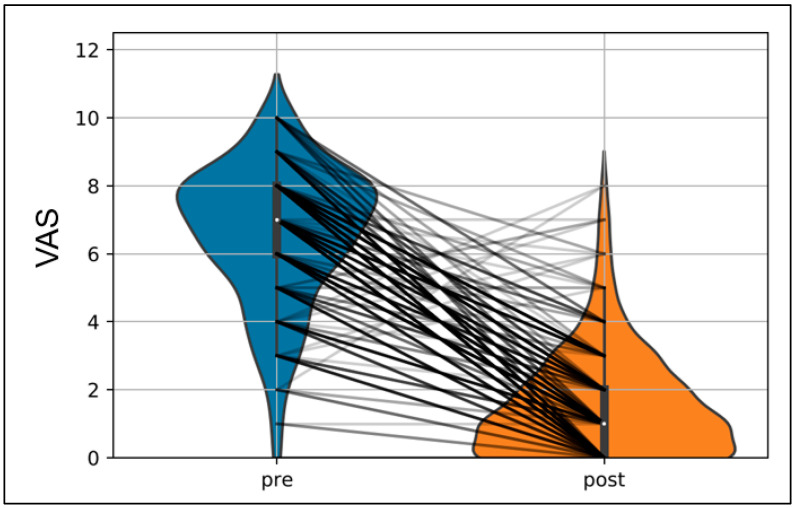
Violin plots of the changes in the subjective rating of pain (visual analog scale (VAS)) at pre- and post-condition (VAS_pre: 6.57 ± 2.05; VAS_post: 1.54 ± 1.60). The black lines represent individual changes in the subjective rating of pain. The color saturation maps the occurring frequency of the respective change in the VAS (black = high frequency; grey = low frequency).

**Table 1 jfmk-08-00066-t001:** Age-related and gender-specific data of the sample.

	Age Groups (Years)				
	<20	20–29	30–39	40–49	>49
Total (n = 340)	80	32	45	44	139
Male (n = 143)	39	10	25	19	50
Female (n = 197)	41	22	20	25	89

## Data Availability

The data presented in this study are available on request from the corresponding authors.

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
