# Peer review of "Assessing the Subjective Effectiveness of Sensorimotor Insoles (SMIs) in Reducing Pain: A Descriptive Multicenter Pilot Study"

_jfmk, 2023, doi:10.3390/jfmk8020066_

Round 1

Reviewer 1 Report

the reviewer apreciated the overall work, with a significant clinical effort (340 people in the cohort, longitudinal fellow-up over 6 months). The work ought to be published  but I suggest some changes in order to have a more balanced point of view.

In fact, too few details are provided on the study design:
Q1- how bias are reduced?
Q2- why the authors choose a one-arm design? How the authors can separate a simple placebo or temporal effect with the insole effect?

In the same way, provided results are interesting, but the reader would like to have more information.
Q3- What about the other results? (e.g. indication, age, gender effects)

Last, the discussion section is very rich, but weakly connected to the results; it tends to be more like a bibliographical discussion. I suggest to synthetize more.
Q4- In any cases, the authors ought to comment on the limitations of the study (including of course the choice of a one-arm design).

Other comments
page 2 line 62: "directional pressure". This vocabulary is not suitable as the pressure is a scalar ad thus directionless
page 2 line 71-72: [23] suggests the mechanism in the discussion part, but the level of proof is very weak considering (1) the number of patients (2) the study design that is not able to seperate proprioceptive and biomechannical effects. May the authors be more cautious in this assessment.

page 4 line 143: Seabor should be Seaborn instead ?
page 5 line 175-176: the question of cause/consequence between pain reduction and joint kinematics is unclear. Even though the authors write "As a result, the corrected foot position and a better foot pressure distribution can lead to a reduction in foot pain", this could also be the case in the other direction.

Reviewer 2 Report

In the following lines a review is provided for the paper "Assessing Subjective Effectiveness of Sensorimotor Insoles (SMIs) in Reducing Pain: A Descriptive Multicenter Pilot Study". 

The issue is very interesting and I want to congratulate with the Authors because it is one of the first paper I ever read underling that the results of the effect of the insoles should be taken carefully into account, although the study sample is very large. Congratulation for the research and for the manner you wrote the paper.

In the following lines some minor changes are suggested to complete the paper, it will be very interesting to read for reasearcher and practitioners, for sure.

1) Lines 37-39: you speak about gait and postural control, it is true. I found on MDPI database this paper about the effect of insoles on gait parameters and I think it is wothy to be insert in your introduction to get a stronger basis: https://www.mdpi.com/1660-4601/17/12/4569 

2) Line 116-117: It is not clear if the VAS scale was specific for foot, achilles tendon, knee and back separately or if the VAS scale was generalized without any specification about the area of pain. It seems the second option, but can it be possible to explicit it better?

3) The detailed description of the sample lacks. It is very useful and it must be added. It is interesting to add information about: health status of the sample, rate of drop-out of the sample (how many subjects completed the study?). Did the orthopedic technicians perform some changes on the given insoles in case of patient's discomfort?

4) The discussion is detailed and well written, in particular the lines 222-262. Congratulation. I suggest to create a sub-paragraph "Limitations" for the lines 263-274.

The conclusion is well written and clear.

Round 2

Reviewer 1 Report

The reviewer thanks the authors for their efforts to answer all the concerns, both in the acompanying letter and in modifying the text.

I truely understand the difficulty of the discussion as it is described, and the lack of well-funded evidence for these devices.